# Longevity is impacted by growth hormone action during early postnatal period

Liou Y Sun[1]*, Yimin Fang[2], Amit Patki[3], Jacob JE Koopman[4], David B Allison[1,3,5], Cristal M Hill[2], Michal M Masternak[6,7], Justin Darcy[2], Jian Wang[1], Samuel McFadden[2], Andrzej Bartke[2]

[1]Department of Biology, University of Alabama at Birmingham, Birmingham, United States; [2]Department of Internal Medicine, Southern Illinois University, School of Medicine, Springfield, United States; [3]Department of Biostatistics, University of Alabama at Birmingham, Birmingham, United States; [4]Section of Gerontology and Geriatrics, Department of Internal Medicine, Leiden University Medical Center, Leiden, Netherlands; [5]Department of Nutrition Sciences, University of Alabama at Birmingham, Birmingham, United States; [6]Burnett School of Biomedical Sciences, College of Medicine, University of Central Florida, Orlando, United States; [7]Department of Head and Neck Surgery, The Greater Poland Cancer Centre, Poznan, Poland

**Abstract** Life-long lack of growth hormone (GH) action can produce remarkable extension of longevity in mice. Here we report that GH treatment limited to a few weeks during development influences the lifespan of long-lived Ames dwarf and normal littermate control mice in a genotype and sex-specific manner. Studies in a separate cohort of Ames dwarf mice show that this short period of the GH exposure during early development produces persistent phenotypic, metabolic and molecular changes that are evident in late adult life. These effects may represent mechanisms responsible for reduced longevity of dwarf mice exposed to GH treatment early in life. Our data suggest that developmental programming of aging importantly contributes to (and perhaps explains) the well documented developmental origins of adult disease.

*For correspondence: leeosun@gmail.com

**Competing interests:** The authors declare that no competing interests exist.

## Introduction

Epidemiological studies of individuals born or conceived during the 'Dutch famine' led to an appreciation of the impact of early-life events on adult health, and this important concept was formalized as the 'Barker hypothesis' (*Barker, 1997*). Low birth weight infants born to undernourished mothers were found to have increased risk for hypertension, cardiovascular disease and diabetes in later life (*Huxley et al., 2002*; *Xiao et al., 2010*). Furthermore, recent studies have shown that maternal overnutrition and obesity are also strongly associated with adult metabolic dysfunction in the offspring (*Curhan et al., 1996*; *Levin and Govek, 1998*). This evidence linked early growth status and nutrient signals to the development of diseases in adult life, and now this concept has been formulated as the Developmental Origins of Health and Diseases (*Hanson and Gluckman, 2014*; *Sinclair et al., 2007*).

It should be noted that the impact of reduced nutrient availability, and the resulting slower growth and development is not always detrimental. Moderate reduction of food or protein intake during pregnancy of female rats and mice lead to improved metabolic status and extended longevity

**eLife digest** For decades, research has shown that early-life events that happen when animals, including humans,are developing as embryos can later influence how those animals look and behave as adults. However, little is known about if the hormones that drive an animal's growth shortly after its birth have long-lasting effects too. For example, do growth hormones influence how quickly an animal will age, how healthy it is during adulthood, and how long it will go on to live?

Sun et al. now show that increasing the levels of growth hormones in young mice for just six weeks can have a long-lasting effect on the animals' lifespans. The experiments involved normal mice and dwarf mice, which are smaller and live for longer. From when they were one week old until they were seven weeks old, the mice were given either growth hormone or salty water as a negative control. As expected, the growth hormone helped the animals to grow longer and heavier. However, Sun et al. also found that this treatment significantly shortened the lives of the male dwarf mice, but not the female dwarf mice or the normal mice. Indeed, male dwarf mice given the growth hormone lived lives that were 20% shorter than those of male dwarf mice given the negative control, and further analyses suggested that they aged faster too.

The biochemical processes that occur within a living animal in order to keep it alive are collectively referred to as the animal's "metabolism". Further experiments showed that the metabolism of adult dwarf mice that had been exposed to growth hormone at a young age was different from the dwarf mice that had been given the negative control instead. These metabolic changes could help to explain why exposure to growth hormone at an early stage in life can affect an animal into its adulthood.

The next step will be to work out, in molecular detail, how exposure to growth hormone in early life has a long-lasting effect on aging and lifespan. Such studies might help scientists to understand more about how a person's experiences during childhood could affect them in later life, including how it affects their risk of developing age-related diseases.

of offspring (*Hales et al., 1996*; *Jennings et al., 1999*; *Ozanne and Hales, 2004*). We have shown that a modest reduction of nutrient availability during the pre-weaning period produced by increasing the number of pups in a litter increased longevity of UM-HET3 mice (*Sun et al., 2009a*). Likely, the mechanisms of the effects of both under- and over-nutrition on adult disease, aging and longevity include alternations in hormonal signaling. Hepatic responsiveness to GH, the resulting changes in plasma IGF-1 levels, insulin levels and insulin sensitivity represent one possible mechanisms. There is considerable evidence that hormonal signals are among key determinants of mammalian longevity, and the role of the somatotropic axis (GH-IGF-I axis) in the control of aging is particularly well documented (*Bartke et al., 2013*). Robust and reproducible extension of both female and male longevity characterizes mice lacking GH (*Brown-Borg et al., 1996*; *Flurkey et al., 2001*), GH receptors (*Coschigano et al., 2003*) or GH-releasing hormone, the key stimulator of GH release (*Sun et al., 2013*). Longevity is also extended in mice with reduced tissue levels of bioavailable IGF-1 (*Conover and Bale, 2007*). Importantly, there is increasing evidence that the somatotropic axis similarly influences longevity in other mammalian species including humans (*Guevara-Aguirre et al., 2011*; *Milman et al., 2016*; *van der Spoel et al., 2016*).

Panici et al. reported that GH (but not thyroxine) treatment during early postnatal period had negative effect on the lifespan of male Ames dwarf mice (*Panici et al., 2010*) implying that the actions of GH during development can alter the trajectory of aging and that regulation of longevity may be difficult to uncouple from growth and adult body size. In contrast, an earlier study showed that early-life GH replacement had no effect on longevity in Snell dwarf mice despite dramatic effects on growth and development (*Vergara et al., 2004*). Notably, Snell dwarf mice share the same endocrine defects with Ames dwarf mice and also live much longer than their normal siblings (*Flurkey et al., 2001*). The discrepancy between the results of these studies is likely due to the difference in the protocols including onset and timing of early-life GH treatments and genetic background. Interestingly, in an earlier study using the dwarf rats, GH therapy during development was shown to extend longevity (*Sonntag et al., 2005*), but the interpretation of these findings was

complicated by the unexpectedly normal lifespan of these mutants. In view of the discrepancies between the available results and the potentially important implications of the impact of GH signaling during development on longevity, we re-examine this issue in both sexes of normal and long-lived mutant mice. To search for mechanisms linking early GH treatment with reduced longevity, a separate cohort of Ames dwarf mice was treated for six weeks with GH or saline starting at the age of one week. When these animals reached adulthood (18 months of age), they were used for metabolic profiling and assessment of the hepatic stress signaling, inflammation gene expression and xenobiotic detoxification pathways. Herein, we show that the relatively brief period of GH treatment during the defined early 'developmental window' can have a dramatic impact on the lifespan and age-associated characteristics.

## Results

### Decreased lifespan in ames dwarf mice by transient exposure to GH during early

To evaluate the effect of early-life GH treatment on longevity, we examined the survival of Ames dwarf (*Prop1*<sup>df/df</sup>) mice and normal littermate control mice employing two treatment protocols in which GH or vehicle (saline) was administered starting at the first or second postnatal weeks.

Consistent with the previous reports (*Bartke et al., 2001*; *Brown-Borg et al., 1996*), saline treated dwarf mice had dramatically extended lifespan relative to N control mice (p<0.0001; log-rank test, both sexes). Being subjected to the GH treatment between the postnatal first and seventh week, the median lifespan of GH treated dwarf mice (sexes combined) was decreased by 165 days (or 16%) relative to that of saline-injected dwarf mice (839 days for GH-dwarf mice vs. 1004 days for saline-dwarf mice) (*Figure 1A*; left panel), (p=0.0382) based upon log-rank test. Intriguingly, there was no statistically significant (log-rank test) effect of the same GH treatment regimen on the lifespan of control mice. Analysis of each sex separately showed that median lifespan in male *Prop1*<sup>df/df</sup> mice was shortened by 204 days (20%; from 1011 to 807 days) by GH treatment between postnatal first and seventh week (*Figure 1B,C*), with the overall survival being significantly decreased (p=0.008). There was no significant treatment effect on overall or median survival in female dwarf mice, indicating sex dimorphism in response to the early GH exposure.

To determine the effect on the maximal longevity, quantile regression method was employed to compare the proportion of live mice in each group at the age at which only upper percentiles (that is, 25th or 10th) of the population remained alive (*Wang et al., 2004*). As shown in the *Figure 1A*, this (weeks 1–7) protocol in dwarf mice led to a significant decrease in maximum lifespan (p=0.0462 for 25th percentile and p=0.0400 for 10th percentile) relative to vehicle controls. Cox Proportional Hazzard (PH) models show that GH treatment mice have significantly higher hazards, compared to Saline, in males except in littermate control mice in week 1 data. *Supplementary file 1* indicates significantly larger Hazzard Ratios ranging from 2.45 to 6.7 for males. Linear regression models show very consistent results with Cox PH model results. As illustrated in *Supplementary file 1*, except for littermate control males, GH treated male mice have significantly lower adjusted mean life span compared to saline group male mice.

In the setting of the second of the employed treatment protocols (GH or vehicle between postnatal second and eighth week), the median lifespan of the GH-treated dwarf mice (sexes combined) was decreased by 199 days (19.5%; from 1019 to 821 days) relative to that of vehicle-treated dwarf mice (log-rank test, p<0.0229). In dwarf mice, the median lifespan was decreased by 22% (p=0.011) in males and 19.6% (p=0.048) in females. Pooled and sex-specific data are summarized in *Supplementary file 1*. In contrast with the results of treatment earlier in life (weeks 1–7), GH treatment between weeks 2 and 8 significantly shortened the longevity of littermate control mice (log-rank test, p=0.0002), with the median lifespan reduced by 12.6%. Littermate male GH-treated mice had a 19.6% decrease in mean lifespan relative to the saline group (*Figure 2*) (log-rank test, p<0.0001) (*Supplementary file 1*). The longevity of GH-treated littermate female mice seemed slightly decreased compared to the saline group, but this apparent difference was not statistically significant (*Figure 2*).

To further evaluate the effects of early-life GH treatment on the aging process, we calculated aging rates per age interval of the dwarf mice and control mice in the two GH treatment protocols.

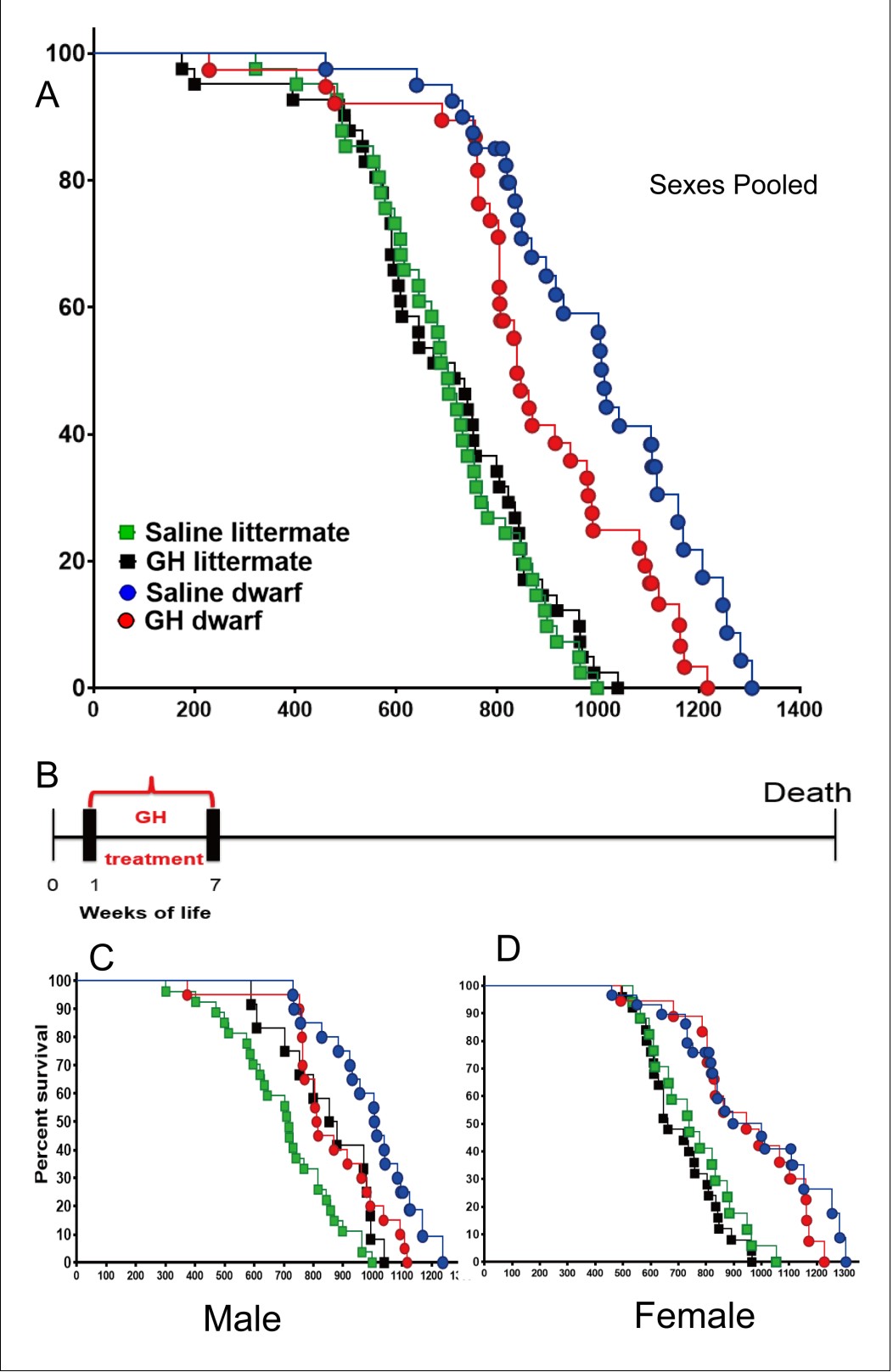

**Figure 1.** Effects of early-life GH treatment on longevity starting at the first postnatal week. (**A**) Sex pooled Kaplan-Meier survival curves for each treatment and genotype: Ames dwarf (*Prop1*df/df) and Littermate control mice treated with either vehicle (saline) or GH; each point represents a single mouse. N = 41 for control mice groups with Saline; N = 41 for control mice groups with GH; N = 31 for dwarf mice groups with Saline and N = 36

*Figure 1 continued on next page*

*Figure 1 continued*

for dwarf mice groups with GH. (B) Experimental scheme detailing administration time of GH and vehicle treatment between postnatal first and seventh week. (C) Male survival curves. (D) Female survival curves.

We employed a non-parametric method, explained and applied previously (*Koopman et al., 2016*). In the first week GH treatment protocol (between postnatal first and seventh week), GH treated dwarf mice had aging rates that increased at younger ages to a lower level relative to saline-treated dwarf mice (*Figure 3A*). There was no clear effect on aging rates of normal control mice using the same GH treatment regime. These patterns are consistent with the effects on lifespan and survival discussed above. In the second week of the GH protocol (between postnatal second and eighth week), a similar pattern was observed in GH treated dwarf mice with increased aging rates at younger ages relative to saline controls (*Figure 3B*). In contrast with first week results, GH treated control mice had aging rates that increased at younger ages but did not reach the same level as the saline treated mice in later life. These patterns are consistent with the effects on lifespan and survival discussed above. Moreover, these effects of GH are consistent with the effect of reduced GH signaling in other mutant mice, which have aging rates that increase at the advanced ages to a higher level relative to control mice (*Koopman et al., 2016*).

Together, the reduction in lifespan of these long-lived *Prop1*$^{df/df}$ mice by early-life GH treatment indicates that alteration in hormonal signaling during critical developmental windows is pivotal for lifespan determination and potential susceptibility to age-related illness.

## Phenotypic characteristics in *Prop1*$^{df/df}$ mice were altered by Early-life GH treatment

As expected, early-life GH treatment led to a significant somatic growth of dwarf mice in comparison to the saline-treated dwarf mice ($p < 0.01$; *Figure 3C and D*). This increase in body weight persisted after the treatment was stopped although the growth almost leveled off, as expected. Notably, significant increase in body length was also observed in GH-treated dwarf mice indicating that the body weight gain was associated with skeletal growth (*Figure 4A*). Moreover, absolute heart, kidney and liver weight were increased in GH-treated dwarf mice compared to the vehicle-injected group (*Figure 4C*; $p < 0.05$) whereas brain weight was not affected (*Figure 4B*). In contrast, the weight of the subcutaneous white adipose tissue (WAT) was dramatically decreased by the GH treatment in dwarf mice (*Figure 4C*).

To search for mechanisms linking early GH treatment with reduced longevity, a separate cohort of Ames dwarf mice was treated for six weeks with GH or saline starting at the age of one week. To further evaluate the long-term metabolic consequences of this early GH treatment, we assessed the metabolic alterations related to energy expenditure (EE) in these mice at the age of 18 months by means of indirect calorimetry. Similar to our previous reports (*Westbrook et al., 2009*), O2 consumption rate ($VO_2$), which declines during normal aging, was found increased in the adult dwarf mice when compared with age-matched littermate control mice (*Figure 4E*). Remarkably, this upregulation of $VO_2$ in 18-month-old dwarf mice was almost completely abolished by the transient early GH exposure (*Figure 4E*). Furthermore, we examined the relative metabolic fuel utilization by measuring the respiratory quotient (RQ), which is a dimensionless ratio comparing the volume of carbon dioxide to the volume of oxygen consumed over a given time ($RQ = VCO_2/VO_2$) (*Johnston et al., 2006*). The values of RQ for mice usually range from 1.0 which means that carbohydrate is the primary source of energy to around 0.7 which means that energy comes from fat oxidation (*Even and Nadkarni, 2012*). The dwarf mice were found to have a lower RQ value than control mice confirming our previous observations (*Westbrook et al., 2009*) while early-life GH treatment significantly increased RQ value (*Figure 4F*). These data together suggest that early-life GH therapy dampens the metabolic flexibility in Ames dwarf mice by reducing their capacity of greater utilization of fat as energy source.

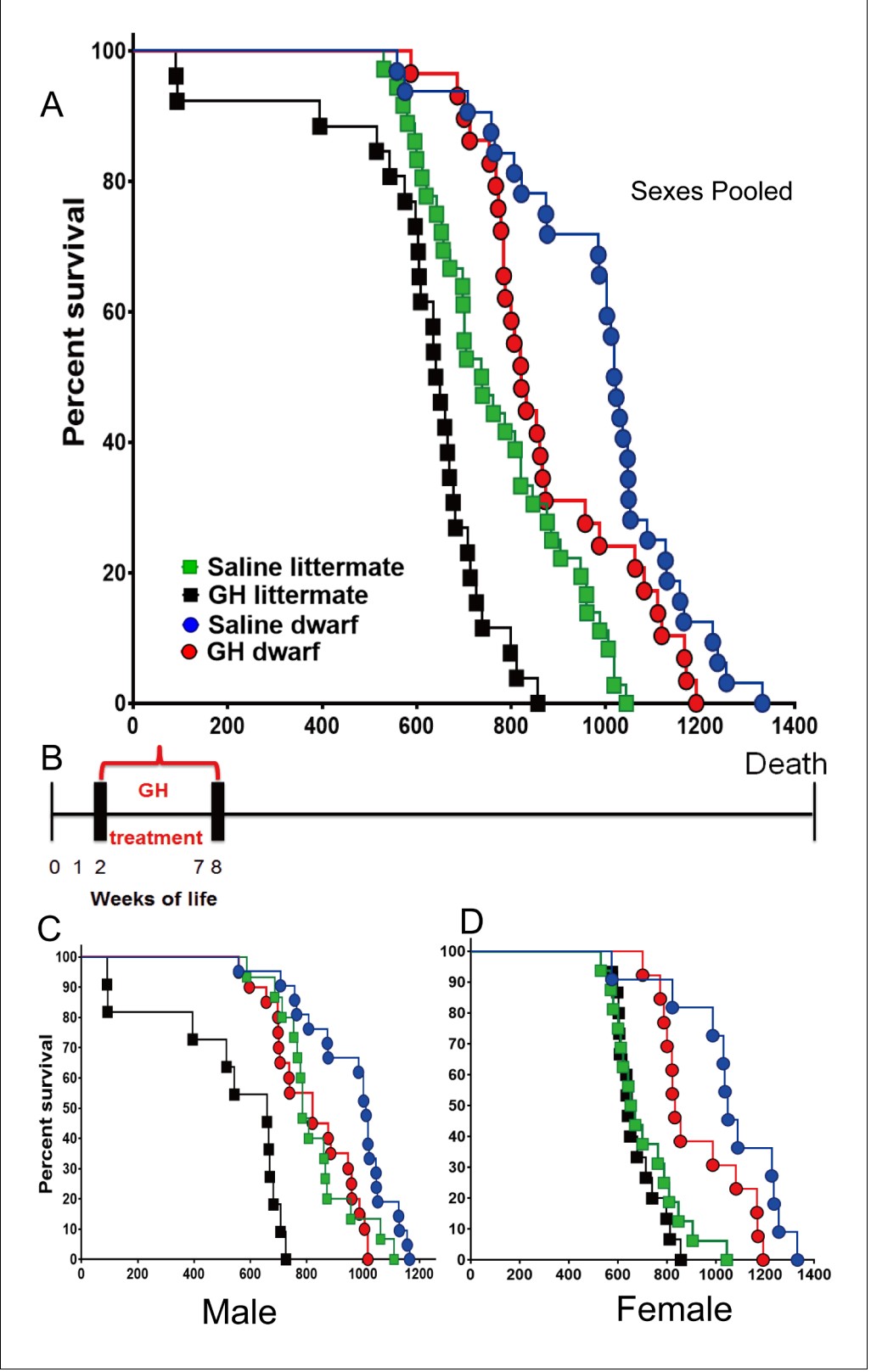

**Figure 2.** Effects of early-life GH treatment on longevity starting at the second postnatal week. (**A**) Sex pooled Kaplan-Meier survival curves for each treatment and genotype: Ames dwarf (df) and Littermate control mice treated with either vehicle (saline) or GH; each point represents a single mouse. N = 36 for control mice groups with Saline; N = 26 for control mice groups with GH; N = 32 for dwarf mice groups with Saline and N = 29 for

*Figure 2 continued on next page*

*Figure 2 continued*
dwarf mice groups with GH. (**B**) Experimental scheme detailing administration time of GH and vehicle treatment between postnatal second and eighth week. (**C**) Male survival curves. (**D**) Female survival curves.

## The effect of Early-life GH on metabolic profile

Improved insulin sensitivity and increased metabolic homeostasis are the shared features of long-lived GH-related mutant mice (*Bartke, 2011*; *Bartke et al., 2013*). As expected, fasting plasma glucose and insulin concentrations were significantly reduced in dwarf mice. Importantly, the early-life GH exposure increased both circulating insulin and glucose to the levels measured in normal littermate controls at the ages of 20 months (*Figure 5A*). Similarly, the adiponectin level, a marker of insulin sensitivity, was higher in dwarf mice than in littermate control mice and GH treatment

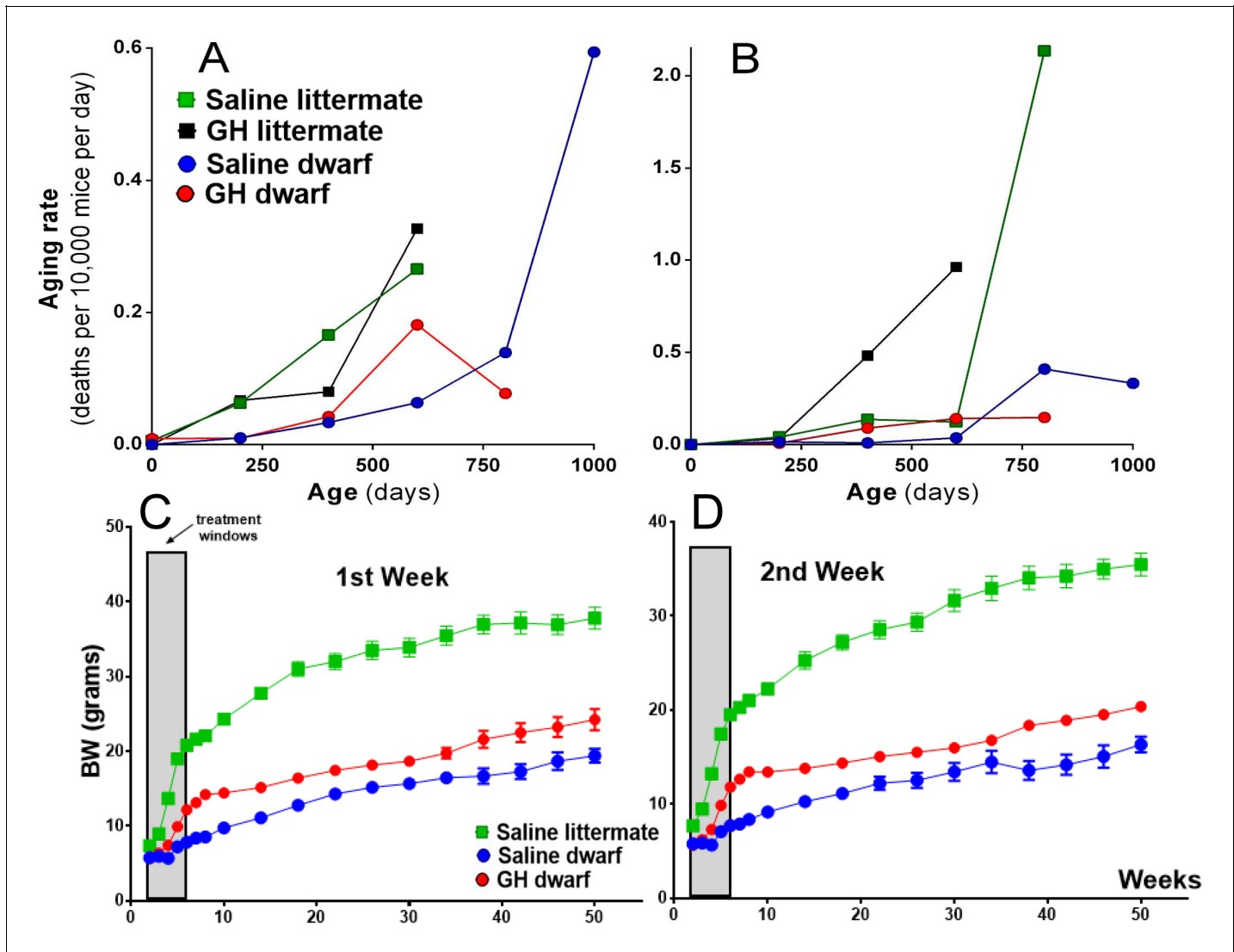

**Figure 3.** Age-dependent aging rates of the groups of mice in the first week (**A**) and the second week (**B**) GH treatment groups. The aging rates describe the increases in the mortality rates with age and are expressed in deaths per 10,000 mice per day, which equals the change in mortality rate per day. Data from both sexes were pooled for each genotype. (**C** and **D**) Body weight of Ames dwarf and control mice subjected to early-life 6 weeks of GH/vehicle treatment. Time points represent mean ± SE weight of each group. (N = 15 mice/group). Data from both sexes are pooled for each genotype.

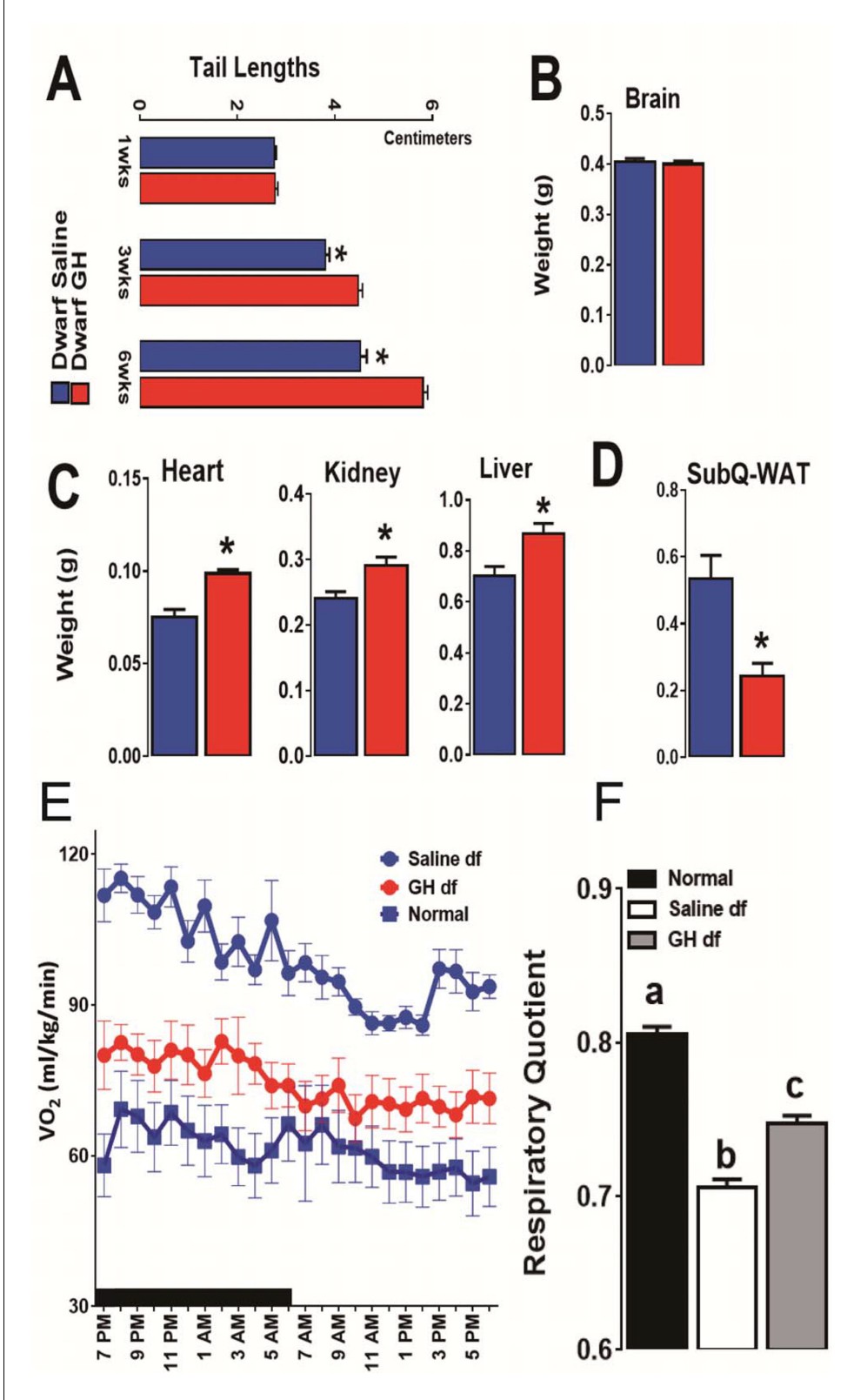

**Figure 4.** Impact of early-life GH treatment on physiological characteristics. (A) Tail lengths; the weights of (B) Brain, (C) Heart, Kidney and Liver and (D) Subcutaneous White Adipose Tissues weights presented as absolute values. (E) $VO_2$ values plotted as hourly averages representing either dark or light periods. (F) RQ = respiratory quotient values. each bar represents means ± SEM for 8–10 mice per group. *p<0.05, **p<0.01, Data from male mice are presented.

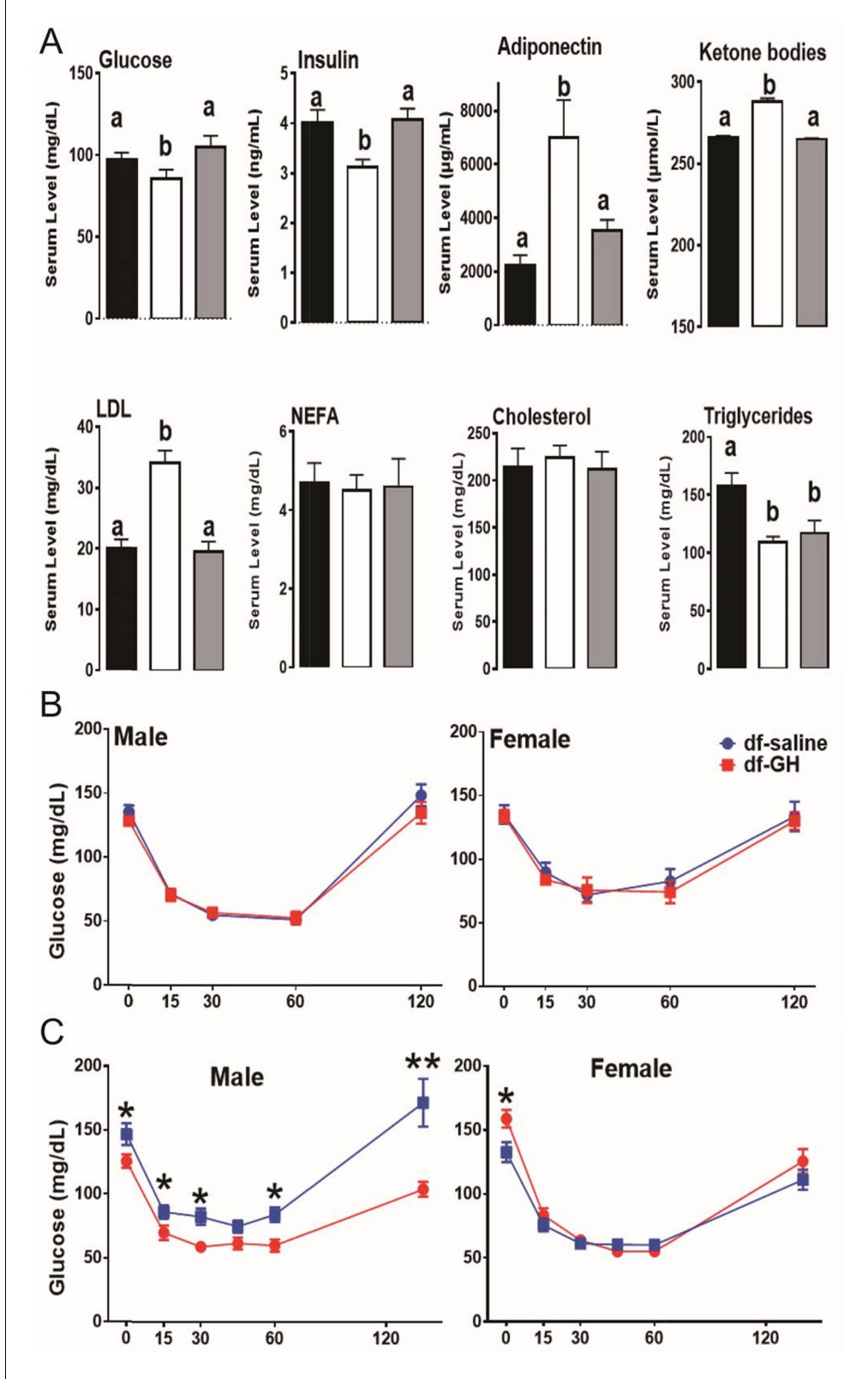

**Figure 5.** Metabolic alterations in responses to early GH treatment. (**A**) Various plasma parameters from male Ames dwarf (*Prop1*df/df) and Littermate control male mice (**N**) subjected to early-life GH treatment. Saline-treated-control mice (Black Bar), Saline-treated-dwarf mice (White Bar) and GH-treated-dwarf mice (Grey Bar), [a,b] values that do not share a superscript letter are statistically significant ($p < 0.05$). Data represent the means ± SEM. Insulin

*Figure 5 continued on next page*

*Figure 5 continued*

tolerance test (ITT) measured at (**B**) 6 months and (**C**) 18 months of age in both male and female mice. Mice were i.p. injected with 1 IU porcine insulin per kg of BW. N = 8 mice per group; *p<0.05.

dramatically suppressed this upregulation (p<0.01; *Figure 5A*). Similar pattern was also observed in the circulating ketone bodies and LDL levels. Interestingly, early GH administration did not affect the plasma triglycerides levels in dwarf mice, and there were no alterations in nonesterified fatty acids (NEFA) or cholesterol level in either genotype (p=0.51).

Further, by employing insulin tolerance test (ITT), we evaluated the sensitivity of blood glucose levels to the action of insulin in a cohort of dwarf mice at 6 months of age. Unexpectedly, both male and female early GH-treated dwarf mice displayed similar sensitivity to insulin as saline-treated groups (*Figure 5B*); with the reduction of blood glucose levels in response to the insulin challenge paralleling between the two groups. We repeated the ITT experiment when dwarf mice were 18-month-old. As seen in *Figure 5B*, male GH-treated dwarf mice exhibited dampened sensitivity to insulin in comparisons to the saline-treated groups (*, p>0.05; at time points measured); whereas female GH-treated dwarf mice had similar response to that measured in the saline groups (*Figure 5B*).

Collectively, we found that a six-week period of early-life GH exposure partially normalized ('rescued') many phenotypic characteristics of dwarf mice. These data further support the notion that the early-life hormonal milieu can induce a long-lasting effect on metabolic homeostasis and healthspan later in life.

## Hepatic stress responsive pathways

Recent evidence has shown that the stress responsive pathways including RAS-MAPK and PI3K-Akt play a critical role in lifespan extension in flies and worms (*Longo and Fabrizio, 2002*; *Slack et al., 2015*). Our previous studies have shown that Erk and Akt signaling are diminished, in response to multiple forms of stress, in cells and tissues from long-lived GH deficient mice (*Sun et al., 2011*, *2009b*). To identify signaling pathways responsible for the decrease of lifespan in Ames dwarf mice by the early-life GH, we first evaluated the effects of this treatment on several protein kinases which were shown to be affected in our previous reports (*Sun et al., 2011*, *2009b*). In contrast to the littermate control mice, dwarf mice had lower hepatic phosphorylation of MAPKs, including the ERK1/2, P38 and Akt kinases, each of which is known to participate in cellular stress responses (*Figure 6A*). However, the livers of GH-treated dwarf mice had elevated levels of phosphorylated EKR1/2, P38 and Akt (ser473) which were indistinguishable from those measured in control mice. Interestingly, Akt phosphorylation on Thr308 was not altered (*Figure 6A*). We next examined the effect of early GH treatment on the transcriptional regulation of immediate early genes (IEGs). IEG including Egr-1, Fra1, Fos, and Jun are rapidly and transiently induced in response to stress or mitogen (*Criswell et al., 2005*; *Sukhatme et al., 1988*). Importantly, Akt and MAPK signaling have been found to regulate IEG induction (*Murphy et al., 2004*). By qRT-PCR, hepatic IEGs mRNA levels were found to be significantly lower in saline-treated dwarf mice than in littermate control mice (*Figure 6B*) whereas early-life GH reversed this suppression and even augmented Egr-1 and Fra-1 mRNA levels above the values detected in control mice. Consistently, hepatic IGF-I mRNA levels, GH sensitive markers, followed a similar pattern (*Figure 6B*).

## Inflammation gene expression

One of the hallmarks of aging is a state of chronic low-grade inflammation in multiple tissues. To determine the early-life GH effects on tissue inflammation status, we examined gene expression of several indices of inflammation in white adipose tissue (WAT), liver, and brain (cerebral cortex) by qRT-PCR. As shown in *Figures 7A* and 20-month-old dwarf mice exhibited lower mRNA levels of several proinflammatory cytokines including IL-6, IL-1$\beta$, MCP-1, TNF-$\alpha$ and Socs3 particularly in WAT and liver. Importantly, early GH treatment of dwarf mice increased expression of the inflammatory cytokines in adipose and hepatic tissues to the level of the age-matched littermate control mice. Interestingly, brain tissues (cortex), in contrast to other tissues, showed no such effect. These data

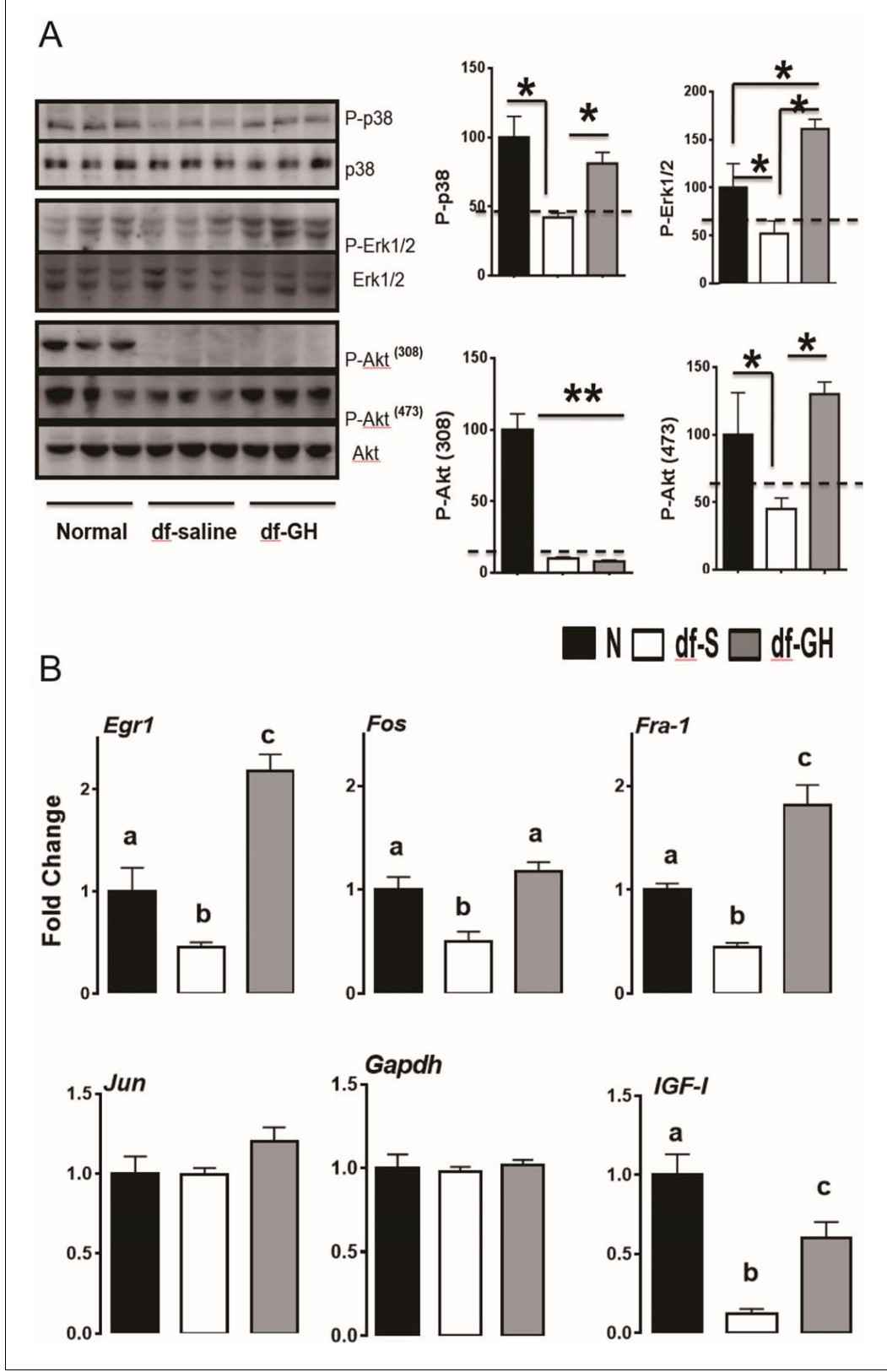

**Figure 6.** Hepatic cellular stress responsive pathways. (**A**) Representative Western blots for phosphorylated and total forms of P38, Erk1/2 and Akt protein in liver lysates of male Ames dwarf (*Prop1*[df/df]) and Littermate control male mice (**N**) subjected to early-life GH or saline treatment. (**B**) Expression of Egr1, Fos, Fra-1 and IGF-I mRNA

*Figure 6 continued on next page*

*Figure 6 continued*

levels in the liver of GH treated dwarf and normal mice. Data are normalized to Gapdh values and expressed as a ratio (fold change) to the level seen in saline injected mice. Saline-treated-control mice (Black Bar), Saline-treated-dwarf mice (White Bar) and GH-treated-dwarf mice (Grey Bar), N = 8 male mice per group; *p<0.05, **p<0.01, ***p<0.001. [a,b] values that do not share a superscript letter are statistically significant (p<0.05).

suggest that GH signals during the early critical period have long-lasting effects on inflammation status in some tissues i.e. WAT and liver but not in other tissues such as cortex. To further understand how the early-life GH regulates tissue inflammation tone, we examined its effect on stress-responsive and metabolic-sensitive signaling molecules in these two tissues. Consistent with the data of inflammatory cytokines expression, GH-treated dwarf mice exhibited elevated hepatic JNK phosphorylation and even a larger increase in NF-kB activation in livers (*Figure 7B*). Moreover, both P-JNK and P-IkB levels were also increased in WAT from GH injected dwarf mice. Together, these data clearly showed that early transient GH exposure influences the inflammatory status in late life in sensitive tissues including WAT and liver.

## Effects on hepatic expression of Xenobiotic detoxification genes

The liver is a major organ to detoxify and eliminate xenobiotics and endobiotics which plays a key role in the metabolic homeostasis of the organism (*Österreicher and Trauner, 2012*). Recent studies in GH-related mutant mice have linked the activation of xenobiotic signaling with delayed aging and increased lifespan (*Amador-Noguez et al., 2007*; *Steinbaugh et al., 2012*). To investigate the effects of early-life GH exposure on the expression of hepatic xenobiotic genes, we measured hepatic mRNA levels of a set of phase I and phase II xenobiotic detoxification genes through qRT-PCR. Consistent with the previous reports (*Amador-Noguez et al., 2004*), the hepatic expression of these xenobiotic genes was greatly upregulated in saline-treated dwarf as compared to the littermate control mice as shown in *Figure 6A* (p<0.001). Early-life GH treatment dramatically suppressed the elevation of these genes including Cyp2b9, Cyp2b13, Hao3, FMO3 and Sth2 (two-tailed t-test; p<0.001) (*Figure 8A*). There was no such effect on Gpadh mRNA, the housekeeping control gene. These data indicate that early GH signaling plays a key role in setting-up the xenobiotic regulation pattern in later life.

Several reports have pointed out that farnesoid X receptor (FXR) is a crucial factor in the regulation of xenobiotic detoxification genes in mice (*Amador-Noguez et al., 2007*; *Lee et al., 2010*). We found that FXR protein level was increased in the dwarf livers (*Figure 8B*) despite the lack of difference in the mRNA levels of FXR between the two genotypes. Interestingly, as shown in *Figure 8B*, early-life GH treatment almost completely suppressed the upregulation of hepatic FXR protein in dwarf mice, indicating that the regulatory impact of the FXR activity on hepatic xenobiotic genes is dependent on GH signaling pathways during early-life.

## Discussion

Epidemiological evidence inspired numerous studies linking early-life nutritional environment to the risks of later obesity, diabetes, hypertension, and cardiovascular diseases (*Hanson and Gluckman, 2014*). The results of these studies raise a question of how the factors and signals during the early life may influence mammalian aging. We believe that this question is of fundamental importance from both individual and public health perspectives. In an attempt to address this issue, in the current study, we have examined the role of GH signaling during development and obtained new evidence that transient GH exposure during pre- and early post-weaning period can remarkably shorten longevity of long-lived hypopituitary mice. The striking effects of GH treatment of juvenile mice on their aging trajectory provide a novel opportunity to investigate the role of early-life hormonal environment in longevity and aging at the mechanistic level.

Although the existence of critical developmental time window is known in other contexts, the effects of the early-life events on aging process have not been well studied. Earlier studies in rodents have shown that reduced growth in utero followed by rapid postnatal catch-up growth shortened lifespan (*Jennings et al., 1999*; *Ozanne and Hales, 2004*). Interestingly, Ozanne *et al.* also found

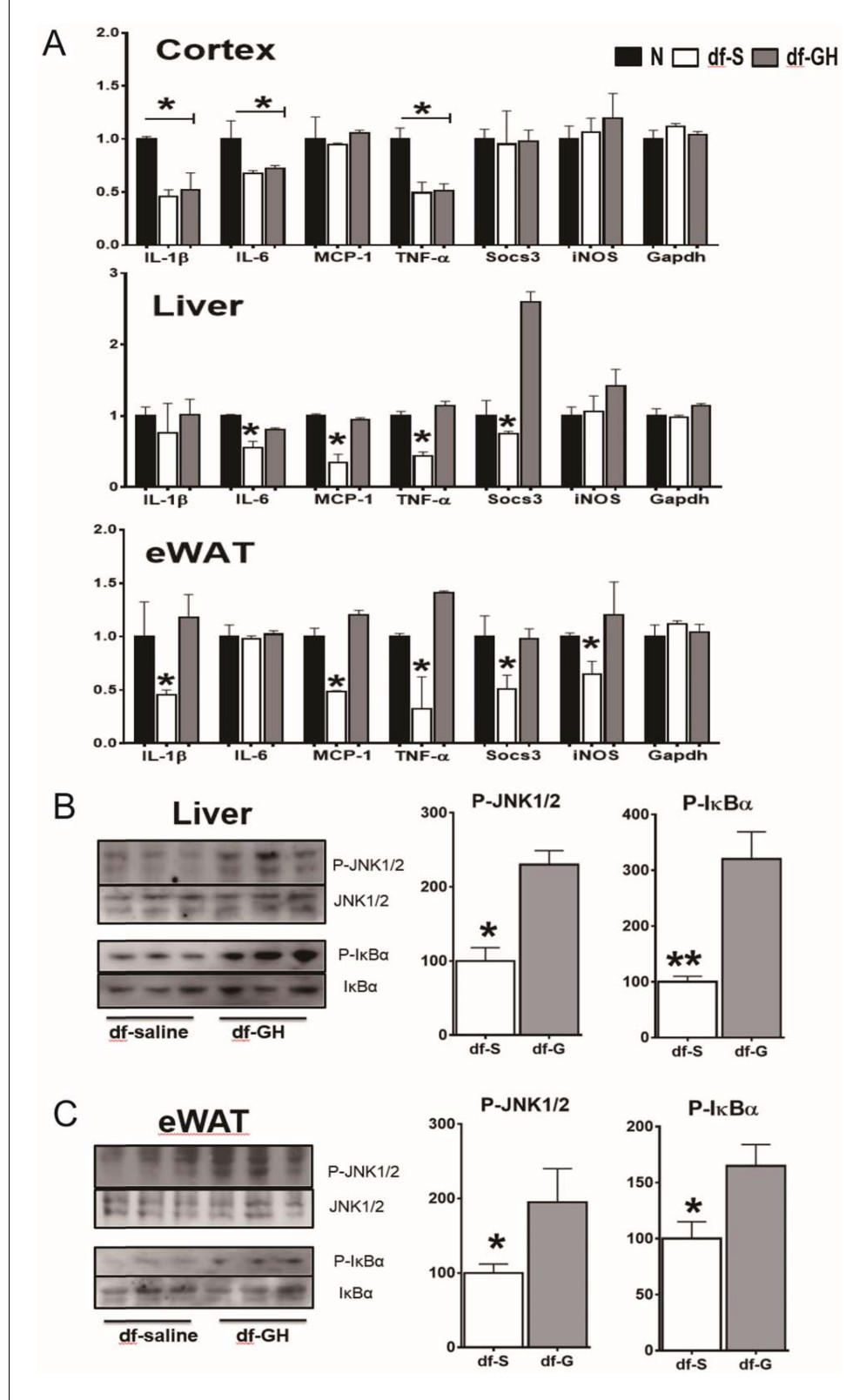

**Figure 7.** Long-term effects on tissue inflammatory markers. (**A**) Effects of early GH treatment on the expression of various inflammatory indices (IL-6, IL-1$\beta$, MCP-1, TNF-$\alpha$, Socs3 and iNOS in WAT, liver, and cerebral cortex by qRT-PCR. Data are normalized to GAPDH or actin values and expressed as a ratio (fold change) to levels of mRNA in control mice. (**B**) Representative Western blots for phosphorylated and total forms of JNK and IkB in liver and

*Figure 7 continued on next page*

*Figure 7 continued*

eWAT lysates of dwarf subjected to GH or saline treatment. N = 8 male mice per group; *p<0.05, **p<0.01, ***p<0.001.

that offspring that were nursed by dams fed a low protein diet, had increased lifespan (*Ozanne and Hales, 2004*). Along with these studies, our previous observations have shown that a 50% increase in litter size limited to the first three weeks of life in genetically heterogeneous male mice resulted in a significant increase of longevity (*Sun et al., 2009a*). Intriguingly, Holzenberger and colleagues reported that alterations of the nutritional status during the suckling period modified the growth trajectories and metabolic plasticity in late life through the regulations of the GHRH-GH-IGF-axis in mice (*Kappeler et al., 2009*). It is noteworthy that a series of studies in rats conducted by Vickers and his colleagues have shown that GH treatment from postnatal day 3 to day 22 could ameliorate or reverse the detrimental consequences of maternal under-nutrition indicating that hormonal intervention during this critical period can override prenatal predispostion (*Gray et al., 2014, 2013*; *Li et al., 2015*; *Reynolds et al., 2013*). A recent report has shown that disrupting the GH signaling during adulthood by induced disruption of the GHR gene did not affect the median lifespan of the mice although it led to a modest increase in the maximal lifespan of the females (*Junnila et al., 2016*). These findings strongly support the critical role of developmental GH signaling in lifespan regulations (*Junnila et al., 2016*). Consistent with these evidences, our study pinpoints the periods between the first and the eighth week of postnatal life as a bona fide critical developmental time window in which longevity and aging rate could be programmed by GH signals. Using GH deficient dwarf mice in which hormonal signaling, growth and adult body size are drastically altered and longevity is greatly increased, our experiments provide compelling evidence that transient hormonal alterations during early critical developmental periods has crucial and long-lasting effect on lifespan.

An important advantage of using these mutant mice and their genetically normal siblings in this type of studies is the opportunity to make direct comparison of phenotypic characteristics in individuals that developed concurrently in the same uterus and were raised together by the same dam in an identical environment and yet have vastly different longevity phenotypes. The observation that early GH treatment can shorten the longevity of littermate control mice is novel and potentially important. It can be interpreted as additional evidence that GH actions during development predispose to accelerating aging. Thus, increasing GH action during early post-natal life leads to a reduction of longevity even if GH signals are not altered after eight weeks of age, which is more than 90% of the animal lifespan. However, these intriguing findings have to be interpreted with caution. Unlike GH deficient dwarf mice, the effect of exogenous GH is complicated by the impact of this treatment on endogenous GH activity and action. It is noteworthy that longevity was shortened by exogenous GH only in N male and in only one of the two experiment protocols.

As indicated in multiple studies of long-lived mice with mutations affecting the somatotropic axis and mice on long-term calorie restriction, decreased insulin and glucose levels together with improved insulin sensitivity are associated with slow-aging most likely mechanistically (*Bartke, 2011*). Notably, these somatotropic mutant mice have markedly increased oxygen consumption ($VO_2$) per unit of body weight and concomitantly reduced respiratory quotient (RQ) (*Westbrook et al., 2009*). Strikingly, our new data have shown that these characteristic features are reversed by a few weeks of exposure to GH during early postnatal period. Our findings suggest that the advantageous metabolic features of these long-lived mice are established during the early postnatal life and are particularly susceptible to the alterations in hormonal environments. Taken together, this evidence suggests that that first few weeks of postnatal life represent a critical window in which the set points related to aging and longevity are programed by the hormonal milieu. Another important implications of the present findings is that it may be difficult or perhaps impossible to completely uncouple the extended longevity of these animals from the reductions in somatic growth and adult body size.

The mechanisms through which aging and longevity is programmed by the early-life GH signals remain unclear. In current studies, we focused on the possible involvement of the stress responsive pathways, inflammation signaling and xenobiotic detoxification gene expression. Our results suggest that the effects of the transient early GH action on gene expression and consequent functional

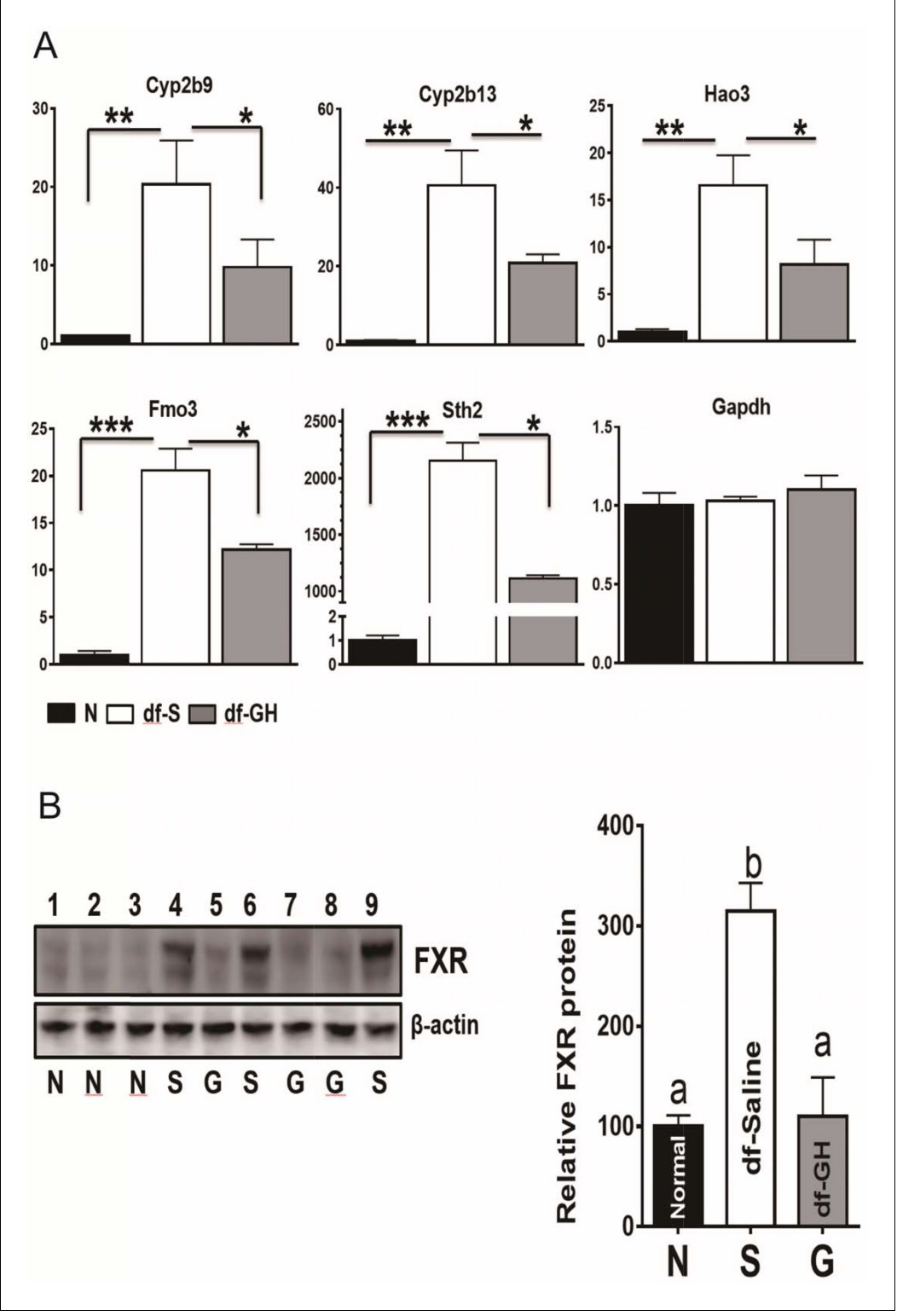

**Figure 8.** Alterations in Xenobiotic Detoxification Genes (XDG) and FXR. (**A**) GH treatment during early-life has a dramatic effects on hepatic XDG expression in male Ames dwarf mice. Typical XDE mRNAs such as Cyp2b9, Cyp2b13, Hao3, FMO3 and Sth2 were measured using real-time RT-PCR. Data are normalized to GAPDH or actin values and expressed as a ratio (fold change) to levels of mRNA in control male mice. (**B**) Representative Western blots for FXR protein in liver lysates of dwarf and control mice subjected to GH or saline treatment. [a,b] values that
*Figure 8 continued on next page*

*Figure 8 continued*

do not share a superscript letter are statistically significant (p<0.05). N = 8 mice per group; *p<0.05, **p<0.01, ***p<0.001.

alterations in various tissues persist into mid- and late adult life and may indeed be permanent. A number of lines of evidence suggest that stress responsive pathways play a critical role in delayed aging and extended longevity (*Miller, 2009*). It is worthwhile to note that long-lived mutant mice were characterized by decreased hepatic phosphorylation of stress related kinases, including the ERK, P38 and Akt, each of which is known to participate in cellular stress responses (*Hsieh and Papaconstantinou, 2006*; *Sun et al., 2011*, *2009b*). In the present study, we have shown that dwarf mice subjected to early postnatal GH administration lost these tissue-specific molecular characteristics, mirroring the change in metabolic features.

Chronic low-grade inflammation contributes to the development of age-related diseases and degenerative pathology during aging. Activation of proinflammatory signaling pathways including JNK and NF-kB activity has been recognized as an important pathophysiological mechanism in mediating the age-associated inflammatory processes (*Cai and Liu, 2012*; *Hirosumi et al., 2002*; *Tuncman et al., 2006*; *Zhang et al., 2013*). In the current study, administration of GH to dwarf mice during the critical developmental windows increased these proinflammatory markers in the livers and WAT indicating activation of JNK and NF-kB kinases in these tissues. In agreement with these observations, Sadagurski et al. found that early-life GH administrations significantly increased or normalized the indices of hypothalamic inflammation in the middle-aged Ames dwarf mice (*Sadagurski et al., 2015*). Together, these data suggest that activation of proinflammatory pathways including JNK and NF-kB may contribute to the loss of metabolic advantages and decrease of longevity in dwarf mice subjected to early-life GH exposure.

Previous studies by us and others suggest that a generalized up-regulation of many hepatic mRNAs for xenobiotic detoxification enzymes (XDE) represents a shared signature of long-lived mice and may be a general mechanism of slow-aging (*Amador-Noguez et al., 2004*; *Shore and Ruvkun, 2013*; *Sun et al., 2013*). The physiology and regulation of hepatic XDEs during early life is not fully understood. Here, in this study, we found that early-life GH administration had a major influence on the elevated expression of these XDE genes in adult Ames dwarf mice. The up-regulation of these typical XDE genes such as Cyp2b9, Cyp2b13, Hao3, FMO3 and Sth2 was dramatically attenuated in the livers from GH-treated dwarf mice. The molecular mechanisms underlying activation of xenobiotic metabolism pathways remain poorly understood. Some studies indicate that the bile acid receptor, also known as farnesoid X receptor (FXR) plays an important role in the regulations of xenobiotic gene expressions (*Amador-Noguez et al., 2007*; *Makishima et al., 1999*; *Pineda Torra et al., 2003*). In the present study, we observed that early-life GH supplement completely abolished the elevation of hepatic FXR protein levels in Ames dwarf mice. This implies that regulation of xenobiotic metabolism pathway by the FXR activity is at least partially mediated by early-life GH signals.

In summary, using GH deficient dwarf mice in which hormonal signaling, growth and adult body size are drastically altered and longevity is greatly increased, our studies provide compelling evidence that transient hormonal alterations during early critical developmental periods has crucial and long-lasting effect on lifespan and longevity-related characteristics. We speculate that mammals during the early development stage respond to their early-life hormonal environment and set anatomical, physiological and biochemical trajectories that shape their future life course including aging and longevity. Hence, understanding the role of hormonal levels and nutrient availability in the first few weeks of postnatal life on preservation of optimal health in old age and mortality risks has potential to make a significant impact on healthcare management and devising interventions that could improve quality of life for older individuals.

## Materials and methods

### Animals and GH treatment

Groups of Ames dwarf ($Prop1^{df/df}$) and littermate control mice (both males and females) were subjected to treatment with porcine GH (pGH) via s.c. injection (6 µg/g bw/d), given in equally divided doses 2×/d starting at the age of 1 or 2 weeks and continuing for 6 wk (dwarf-GH treated). On Saturdays and Sundays, animals were injected only once with a full dosage following a previous protocol (*Panici et al., 2010*). In our study, normal siblings (df/+ or N) of Ames dwarf mice were used as controls. These control mice are heterozygous for the Prop1$^{df}$ mutation and phenotypically not distinguishable from wild type mice. After 6 wk of treatment, the animals were set aside for a longevity study, and mice in these groups were not exposed to any other manipulations except for recording of body weight. Separate cohorts of animals produced using the same protocol were evaluated for metabolic measurements at 6 months of age, for additional metabolic measurements starting at 18 months of age and euthanized around 20 months of age for tissue collections. Animal protocols were approved by the Animal Care and Use Committee of Southern Illinois University and the University of Alabama at Birmingham.

### Aging rates

Aging rates were calculated non-parametrically as described previously (*Koopman et al., 2016*). First, mortality rates were calculated per group of mice per age interval of 200 days by dividing the number of mice that died by the number of days lived by all mice in the age interval of interest. If only one mouse died in the last age interval, this age interval was excluded. Aging rates were derived from the mortality rates per group of mice. The aging rate in each age interval was calculated as the absolute difference between the mortality rate in this age interval and the mortality rate in the subsequent age interval divided by the difference in age between both age intervals.

### Assessment of blood chemistry

Plasma was obtained from blood collected by cardiac puncture from isoflurane anesthetized animals at sacrifice and used for measurement of insulin using Mouse Insulin ELISA Kits (Crystal Chem, Downers Grove, IL). Following the manufacturer's protocol, total ketone bodies and non-esterified free fatty acids (NEFA) were measured using colorimetric assays from Wako Chemicals (Richmond, VA); glycerol was measured using kits from Sigma and triglycerides using kits from Pointe Scientific (Canton, MI), respectively. Adiponectin and resistin levels were assayed using Mouse Adiponectin/Resistin ELISA Kits (Linco Research, St. Charles, MO). Leptin levels were evaluated using Mouse Leptin ELISA Kits (Crystal Chem Inc., Downers Grove, IL). TNF-α and IL-6 were measured using Mouse TNF-α/IL-6 ELISA Kits (Biosource, Camarillo, CA). Plasma FFAs were assayed using optimized enzymatic colorimetric assays (Roche, Penzberg, Germany). Blood was taken from the tail to measure blood glucose using a glucometer (AgaMatrix, Salem, NH).

### Glucose Tolerance Test and Insulin Tolerance Test

Mice fasted for 16 hr underwent GTT by i.p. injection with 1 g of glucose per kg of body weight (BW). Blood glucose levels were measured at 0, 15, 30, 45, 60, and 120 min using a PRESTO glucometer (AgaMatrix, Salem, NH) for GTT. Non-fasted mice were injected i.p. with 1 IU porcine insulin (Sigma, St. Louis, MO) per kg of BW. Blood glucose levels were measured at 0, 15, 30, and 60 min for ITT. The data for ITT are presented as a percentage of baseline glucose. P values were calculated by unpaired, two-tailed Student's t-tests to compare the specific time points.

### Indirect calorimetry

Mice were subjected to indirect calorimetry (PhysioScan Metabolic System from AccuScan Instruments, Columbus, OH) as described before (*Westbrook et al., 2009*). This system uses zirconia, infrared sensors and light beams arrays to monitor oxygen ($VO_2$), carbon dioxide ($VCO_2$), and spontaneous locomotor activity, inside respiratory chambers in which individual mice were tested. All comparisons are based on animals studied simultaneously in eight different chambers connected to the same $O_2$, $CO_2$ and light beam sensors in an effort to minimize the effect of environmental variations and calibration on data. After a 24 hr acclimation period, mice were monitored in the

metabolic chambers for 24 hr with ad libitum access to standard chow (Laboratory Diet 5001) and water. Gas samples were collected and analyzed every 5 min per animal, and the data were averaged for each hour.

## Real-time RT-PCR

Quantitative real-time PCR was performed using a Rotor-Gene 3000 system (Corbett Research, San Francisco, CA) with a QuantiTect SYBR Green RT-PCR kit (Biorad) as described (Sun et al., 2011). In brief, tissues were homogenized with RNA extraction buffer (TRIZOL reagent; Life Technologies, CA) to yield total RNA following the manufacturer's instructions. Total RNA was reverse transcribed with poly-dT oligodeoxynucleotide and SuperScript II. After an initial denaturation step (95°C for 90 s), amplification was performed over 40–45 cycles of denaturation (95°C for 10 s), annealing (60°C for 5 s), and elongation (72°C for 13 s). Amplification was monitored by measuring the fluorometric intensity of SYBR Green I at the end of each elongation phase. Glyceraldehyde-3-phosphate dehydrogenase (GAPDH) or beta-actin expression was quantified to normalize the amount of cDNA in each sample. The change in threshold cycle number ($\Delta Ct$) was normalized to the GAPDH reference gene by subtracting $\Delta Ct_{GAPDH}$ from $\Delta Ct_{gene}$. The effect of treatment ($\Delta\Delta Ct$) was calculated by subtracting $\Delta Ct_{normal}$ from $\Delta Ct_{Tg}$. Fold induction was determined by calculating $2^{\Delta\Delta Ct}$.

## Western blot analysis

Tissues were homogenized in 0.5 ml ice-cold T-PER tissue protein extraction buffer (Thermo Scientific, Rockford, IL) with protease and phosphatase inhibitors (Sigma, St. Louis, MO). 40 µg of total protein were separated electrophoretically according to size by SDS–polyacrylamide gel electrophoresis using Criterion XT Precast Gel (Bio-Rad, Hercules, CA), and blotted with the antibodies. For visualization of specific bands in the chemiluminescence assays, the membrane was exposed to X-OMAT film; for chemifluorescence the membrane was incubated with ECF (enhanced chemifluorescence) substrate and a digital image was generated with the Molecular Dynamics Storm system. Quantification of immunoblot signals was performed using the ImageQuant software package (Molecular Dynamics, Sunnyvale, CA). The following antibodies were obtained for immunoblotting: p38 MAPK, phospho-p38 MAPK (Thr180/Tyr182), ERK, phospho-ERK (Thr202/Tyr204), JNK, phospho-JNK (Thr183/Tyr185), phospho-Akt (Ser473) and Akt, from Cell Signaling Technology (Beverly, MA); Nrf2 antibody from Novus Biologicals (Littleton, CO); $\beta$-actin, inhibitor from Sigma-Aldrich Corp.; and goat anti-rabbit and goat anti-mouse antibodies from Santa Cruz Biotechnology, Inc. (Santa Cruz, CA).

## Statistical analyses

Statistical analyses were performed to test the effects of treatments on lifespan of animals, separately in N and dwarf groups. To examine the effect of treatment on lifespan in each gender, the analyses were done separately for males and females, and also on combined data. Cox proportional hazards survival models were fitted to compare hazards of treatments in each group, both separately and combined by gender. As a part of survival analysis, in addition to Cox proportional hazards survival models, log-rank tests were also performed to test equivalent hypotheses. To assess the effect of treatment on maximum lifespan using quantile regression, Fisher's Exact test was implemented as described previously (Wang et al., 2004). Since there were no censored values in our data, the associations between treatment and lifespan were also tested by fitting linear regression models in addition to survival models. Sex was used as a covariate when data were analyzed combined for males and females.

## Acknowledgements

We thank Adam Spong, Jake Boehm, and Joshua Huber for technical assistance. We also thank other members of the Sun lab for their helpful discussion and Steven Austad for comments on the revision of the manuscript. This work was supported in part by NIH grants AG048264, AG050531 (L Sun), AG019899, AG038850 (A Bartke), P30AG050886, R01AG043972 and P30DK056336 (DB Allison).

## Additional information

### Funding

| Funder | Grant reference number | Author |
|---|---|---|
| National Institute on Aging | | Liou Y Sun |
| National Institutes of Health | AG048264 | Liou Y Sun |
| National Institutes of Health | AG050531 | Liou Y Sun |
| National Institutes of Health | P30AG050886 | David B Allison |
| National Institutes of Health | R01AG043972 | David B Allison |
| National Institutes of Health | P30DK056336 | David B Allison |
| National Institutes of Health | AG019899 | Andrzej Bartke |
| National Institutes of Health | AG038850 | Andrzej Bartke |

The funders had no role in study design, data collection and interpretation, or the decision to submit the work for publication.

### Author contributions

LYS, Conceptualization, Resources, Data curation, Formal analysis, Supervision, Funding acquisition, Validation, Investigation, Visualization, Methodology, Writing—original draft, Project administration, Writing—review and editing, Set up and performed the longevity study; Performed the molecular and biochemical experiments; Performed metabolic studies; YF, Data curation, Methodology, Set up and performed the longevity study; Performed metabolic studies; AP, Resources, Data curation, Validation, Methodology, Writing—original draft, Performed the statistical analyses; JJEK, Resources, Data curation, Software, Formal analysis, Validation, Writing—original draft, Performed the statistical analyses; DBA, Resources, Data curation, Software, Funding acquisition, Validation, Writing—original draft, Performed the statistical analyses; CMH, Data curation, Methodology, Set up and performed the longevity study; performed the molecular and biochemical experiments; performed metabolic studies; MMM, Methodology, Set up and performed the longevity study; JD, Methodology, Performed metabolic studies; JW, Data curation, Formal analysis, Validation, Investigation, Methodology, Performed the molecular and biochemical experiments. Performed the statistical analyses; SM, Methodology, Performed the molecular and biochemical experiments; AB, Conceptualization, Supervision, Funding acquisition, Writing—original draft, Project administration, Set up and performed the longevity study

### Author ORCIDs

Liou Y Sun, http://orcid.org/0000-0002-9802-6780

### Ethics

Animal experimentation: This study was performed in strict accordance with the recommendations in the Guide for the Care and Use of Laboratory Animals of the National Institutes of Health. All of the animals were handled according to approved institutional animal care and use committee (IACUC) protocols (IACUC-20292) of the University of Alabama and (#178-020-001) of SIU school of medicine. The protocol was approved by the Committee on the Ethics of Animal Experiments of the UAB and SIUSOM.

## Additional files

### Supplementary files

• Supplementary file 1. Summary of survival curves and statistical information.

• Supplementary file 2. SAS output of GLM results.

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
