## [Decision Letter]

Thank you for submitting your article "Longevity is impacted by growth hormone action during early postnatal period" for consideration by *eLife*. Your article has been favorably evaluated by Sean Morrison (Senior Editor) and three reviewers, one of whom, Andrew Dillin (Reviewer #1), is a member of our Board of Reviewing Editors. The following individual involved in review of your submission has agreed to reveal their identity: Celine Riera (Reviewer #2).

Overview:

This paper provides unequivocal evidence that altering GH levels during early development, and that even small differences in GH can dramatically alter lifespan and growth. We think that this paper is important because it answers a major question in the aging field. Alterations of insulin/insulin-like signaling (IIS) components that result in dampened insulin signaling promote longevity in many model organisms. However, in mouse models, many of these long-lived animals present growth delay and defects in anterior pituitary function. A burning question in this field remains: Can the long lifespan of these animals be uncoupled from the growth defect due to reduced IIS in some tissues? Therefore, if we rescue the growth defect in Ames mice, can they still live long and present improved metabolic characteristics? Here, it appears that the lifespan extension of this particular reduced IIS model cannot be uncoupled from its growth delay or hormonal imbalances. In our opinion, the paper should be centered around this question.

Specific Concerns:

1) Need to normalize to lean mass. For example, GH supplementation increases body weight and therefore tissue mass. In order to study energy expenditure, the RQ data must be normalized to lean body mass. A large mouse will have reduced energy expenditure compared to a smaller mouse, without necessarily any thermogenic difference. By normalising to lean mass, it will be possible to determine whether the lean mouse actually shows increased thermogenic capacity.

2) Are all the physiological effects sex specific? For many, data from males and females were combined. Given the sex-specificity of lifespan and the phenotypes for which sex was examined, this would be interesting to know. It is not clear in all of the figures whether sexes are combined or not, making interpretation difficult. Can the authors please separate their analysis?

3) No Statistical analysis is provided for mean lifespans.

4) It would be very interesting to know whether this brief period of GH treatment during development changed the time of growth to adulthood. Did it? The authors should report this, and they may wish to comment. Is this hormonal system part of the mechanisms that produces this striking (though not universal) correlation in nature?

5) In many of the figures (Figure 4–Figure 8) it was not clear when/at what age the measurement were made and at what distance from the last GH injection. Maybe it is in the manuscript somewhere but if so, it is hard to find. This is very important to be able to assess the significance of the figures and also of the paper. For example, in Figure 6, IGF-1 levels are much higher in df-GH compared to df-S mice. If this occurs right after the GH injection it would be very much expected, but if this occurs months later it could completely change the significance of the results and of the paper, since it would suggest that the early GH treatment has long-lasting effects on the GH-IGF-1 axis. Thus, GH/IGF-1 signaling later in life may also explain the negative effects of GH exposure only early in life.

Missing Data Files: Lean mass and fat mass measurement are missing.

---

## [Author Response]

Overview:

This paper provides unequivocal evidence that altering GH levels during early development, and that even small differences in GH can dramatically alter lifespan and growth. We think that this paper is important because it answers a major question in the aging field. Alterations of insulin/insulin-like signaling (IIS) components that result in dampened insulin signaling promote longevity in many model organisms. However, in mouse models, many of these long-lived animals present growth delay and defects in anterior pituitary function. A burning question in this field remains: Can the long lifespan of these animals be uncoupled from the growth defect due to reduced IIS in some tissues? Therefore, if we rescue the growth defect in Ames mice, can they still live long and present improved metabolic characteristics? Here, it appears that the lifespan extension of this particular reduced IIS model cannot be uncoupled from its growth delay or hormonal imbalances. In our opinion, the paper should be centered around this question.

We thank the reviewers for their assessment that our work is of interest and important. We certainly agree that addressing the question of coupling/uncoupling of the longevity from the growth delay in these long-lived GH mutant mice is critical. However, our data so far do not provide convincing answer to this question. Indeed, adult body weight of Ames dwarf mice was increased by early GH treatment only slightly (about 15% increase at 1 year of age than saline treated dwarf mice). But median longevity was shortened quite drastically (50% shorter than saline treated dwarf mice) but these mice still live much longer than normal control mice. This evidence may actually indicate ‘partial uncoupling’ between the longevity and growth defects. Nevertheless, we have modified the text (in the last paragraph of the Introduction and in the fourth paragraph of the Discussion) to highlight this critical question and (recognizing comments from reviewers) to integrate this idea better into the manuscript as a whole.

Specific Concerns:

1) Need to normalize to lean mass. For example, GH supplementation increases body weight and therefore tissue mass. In order to study energy expenditure, the RQ data must be normalized to lean body mass. A large mouse will have reduced energy expenditure compared to a smaller mouse, without necessarily any thermogenic difference. By normalising to lean mass, it will be possible to determine whether the lean mouse actually shows increased thermogenic capacity.

This is an important point. Many investigators attempt to normalize energy expenditure (EE) by total body weight (Tschop et al., 2011) however it has been suggested that dividing by the fat-free or lean mass is a preferable approach (Butler and Kozak, 2010; Tschop et al., 2011). Applying this scaling approach to this study is complicated by known differences in body composition in long-lived GH related mutant mice that allometric scaling does not account for. For example, the use of lean mass in the calculations is based on the assumption that smaller animals are leaner. However, Ames dwarf mice have a roughly 40% increase in percent body fat, a 85% increase in relative brain weight, a 30% reduction in relative liver weight, and a nonsignificant reduction in percent lean body mass (Bartke et al., 2013; Bartke and Westbrook, 2012). Thus, the actual metabolic body weight of the dwarf mice may be smaller than a simple scaling exponent would predict. This is further complicated by the fact that adipose tissue in these long-lived mutant mice is very metabolically active (Bartke et al., 2013; Berryman et al., 2004; Darcy et al., 2016). Thus, the allometric scaling exponents will likely to be overestimating the actual metabolic body weight in GH mutant mice and underestimating metabolic body weight in normal animals. When the actual measurements of lean body mass were used, significantly increased VO_2_ was observed in mutant mice compared with control mice in our previous published study (Westbrook et al., 2009). The increases in VO_2_ are of the same magnitude (1.3-fold) as those observed when expressing VO_2_ per gram, indicating that in comparisons of GH mutant with normal control mice the isometric analysis more accurately corresponds to metabolic rate per functional metabolic mass than the allometric scaling exponent estimate.

Unfortunately, we do not have accurate in-vivo body composition analysis by Q-NMR or DXA measurements of lean or fat mass of the Ames dwarf and control mice used in the present study. However, our unpublished data has shown that age-matched Ames dwarf mice have roughly 30% body fat, whereas normal controls have about 20%.

2) Are all the physiological effects sex specific? For many, data from males and females were combined. Given the sex-specificity of lifespan and the phenotypes for which sex was examined, this would be interesting to know. It is not clear in all of the figures whether sexes are combined or not, making interpretation difficult. Can the authors please separate their analysis?

We apologize for this. We have now added an extra sentence to each figure legend to clarify the data of which sex was examined.

3) No Statistical analysis is provided for mean lifespans.

To address this issue, we have re-analyzed lifespan data to test effect of group on mean lifespan using Generalized Linear Models (GLM). We tested the hypotheses of equal mean lifespans between groups “Normal” and “Dwarf” after adjusting for sex where necessary. These results are now highlighted in Table 1 and [Supplementary-material SD1-data]. We are also providing detailed SAS output of GLM results in the [Supplementary-material SD2-data].

4) It would be very interesting to know whether this brief period of GH treatment during development changed the time of growth to adulthood. Did it? The authors should report this, and they may wish to comment. Is this hormonal system part of the mechanisms that produces this striking (though not universal) correlation in nature?

The reviewer raises an excellent point. Unfortunately, we did not determine the age of sexual maturation in the mice of the present study. From our earlier work (Bartke, 1964, 1965) and from the studies of Vergara et al. in Snell dwarf mice (Vergara et al., 2004), we can assume that sexual maturation was advanced by the early GH treatment. In addition, from Figure 3), the growth rate of GH treated dwarf mice has been dramatically slowed down when the GH injections were stopped. And then their growth rate appeared to be parallel to the growth of saline injected dwarfs (Figure 3).

5) In many of the figures (Figure 4–Figure 8) it was not clear when/at what age the measurement were made and at what distance from the last GH injection. Maybe it is in the manuscript somewhere but if so, it is hard to find. This is very important to be able to assess the significance of the figures and also of the paper. For example, in Figure 6, IGF-1 levels are much higher in df-GH compared to df-S mice. If this occurs right after the GH injection it would be very much expected, but if this occurs months later it could completely change the significance of the results and of the paper, since it would suggest that the early GH treatment has long-lasting effects on the GH-IGF-1 axis. Thus, GH/IGF-1 signaling later in life may also explain the negative effects of GH exposure only early in life.

We now make it explicit that we are referring to the findings in the experimental conditions: Separate cohorts of animals produced using the same protocol were evaluated for metabolic measurements at 6 months of age, for additional metabolic measurements starting at 18 months of age and euthanized around 20 months of age for tissue collections. This is now stated in the Materials and methods section.

We appreciate the insightful comment of the reviewers about the hepatic IGF-I mRNA level in Figure 6. Indeed, we have no further data regarding the apparently long-lasting effects on the status of somatotropic axis in late life since only hepatic IGF-I mRNA has been examined. Intriguingly, our previous study showed that IGF-I mRNA was significantly influenced by early-hormonal alterations in a tissue-specific manner (Do et al., 2015). Moreover, Holzenberger’s group reported that changes of GH/IGF-I axis in adults could be directly related to nutrient supply during the early postnatal period (Kappeler et al., 2009). The molecular details concerning how early GH impacts somatotropic axis later in life are currently under investigation and, we believe, outside the scope of this current manuscript.